# Isothermal Hydrogen Reduction of a Lime-Added Bauxite Residue Agglomerate at Elevated Temperatures for Iron and Alumina Recovery

**DOI:** 10.3390/ma15176012

**Published:** 2022-08-31

**Authors:** Olivia Bogen Skibelid, Sander Ose Velle, Frida Vollan, Casper Van der Eijk, Arman Hoseinpur-Kermani, Jafar Safarian

**Affiliations:** 1Department of Materials Science and Engineering, Norwegian University of Science and Technology (NTNU), Alfred Getz Vei 2, 7034 Trondheim, Norway; 2SINTEF Industry, Torgarden, 7465 Trondheim, Norway

**Keywords:** bauxite residue, valorization, hydrogen reduction, iron, leaching, alumina recovery, grey mud

## Abstract

The hydrogen reduction of bauxite residue lime pellets at elevated temperatures was carried out to recover iron and alumina from the bauxite residue in a new process route. Prior to the H_2_ reduction, oxide pellets were initially prepared via the mixing of an industrial bauxite residue with fine calcite powder followed by calcination and high-temperature sintering. The chemical, compositional, and microstructural properties of both oxide and reduced pellets were studied by advanced characterization techniques. It was found that iron in the oxide pellets is mainly in the form of brownmillerite, and calcium–iron–titanate phases, while upon reduction they are converted to wüstite and shulamitite intermediate phases and further to metallic iron. Moreover, it was found that the reduction at lower temperature of 1000 °C is faster than that at higher temperatures of 1100 °C and 1200 °C. The slower rate and extent of reduction at the higher temperatures is attributed to the porosity loss and reduction mechanism change to a diffusion-controlled process step. In addition, it was found that Al-containing phases in the raw materials are converted mainly to gehlenite in sintered pellets and further to the leachable mayenite phase. The alkaline leaching of selected reduced pellets by a sodium carbonate solution yielded up to 87% Al recovery into the solution, while the metallic iron was not affected.

## 1. Introduction

The present commercial process to extract and produce alumina from bauxite ore is the Bayer Process. In addition to producing CO_2_ and being generally energy consuming, it also produces 170 million tons of red mud worldwide annually, per 2019 [1,2]. Depending on the quality of the mined bauxite ore, up to two times red mud is produced per mass of aluminum produced [3]. Red mud is the main by-product of the Bayer Process and there are severe environmental issues concerning its disposal. The most practiced way to dispose of this very alkaline and heavy-metal-containing waste is to store it in ponds around the world. The ponds percolate into the local environments and can potentially flood large areas. Its alkalinity makes it damaging to agriculture and life, depending on groundwater, which threatens the ecosystems surrounding the deposits. Despite being hazardous, the red mud contains considerable amounts of useful minerals and metals such as iron, remnants of aluminum and rare earth elements. Red mud can contain up to 50 precent iron oxides and 10 percent aluminum oxides [4]. These perfectly good raw materials are simply dumped in large deposits—a glaring example of how the linear economy and short-term cycles of supply and demand disregard long-term planning for conservation of natural resources for future generations [4]. Because of the environmental challenges of the red mud, several solutions have been sought out in the past few years [5,6,7,8,9,10,11,12,13,14,15,16], and some of these are based on the Pedersen Process, which targets the prevention of red mud generation. The most important difference of the Pedersen process with the Bayer process is that it produces grey mud, a usable by-product, as opposed to red mud. The main by-product in the Pedersen Process is pig iron, which can further be used in foundries and steelmaking. It also produces CO_2_ in the early stages of the process, although it is reused in later stages, which results in lower overall emissions [13]. Even though the Pederson Process seems to be more eco-friendly than the Bayer Process, it is probably economically inferior and is not used commercially for this reason. However, due to the needs for sustainable development, the process is again under consideration to be revived in a modern form. The ENSUREAL Project is an example of previous research further exploring the use of the Pedersen Process. The project’s main objective is ensuring zero waste production of alumina in Europe. This optimalization of the Pederson Process also include extracting rare earth elements [17].

The reduction of iron with hydrogen has been observed in recent studies [18,19,20]. In bauxite, iron is mainly found in the oxide form called hematite (Fe_2_O_3_), while lower quantity of goethite, FeOOH, may co-exist. Throughout the reduction of the iron oxide with hydrogen, it will form magnetite (Fe_3_O_4_) before further reduction to metallic iron [5]. Depending on the reduction temperature, the process forms different intermediate oxides. When reduction is performed at temperatures above 570 °C and below 1400 °C, it will form wüstite (FeO) before the metallic iron formation [18]. When intending to increase the formation of metallic iron, the oxygen amount should be below 23.25% and above the initial wüstite formation temperature [18]. The hydrogen reduction reactions are shown in reactions (1)–(3), with coherent change in enthalpy at 700 °C [12]. The first reaction explains the formation of magnetite from hematite, while the two subsequent reactions explain the further reduction of magnetite to wüstite, and furthermore, wüstite is reduced to metallic iron.
(1)$$3Fe_{2}O_{3\left(s\right)}+H_{2\left(g\right)}\to 2Fe_{3}O_{4\left(s\right)}+H_{2}O_{\left(g\right)},\;∆H_{700\;{}^{\circ}C}=5279\;\mathrm{kJ}$$
(2)$$Fe_{3}O_{4\left(s\right)}+H_{2\left(g\right)}\to 3FeO_{\left(s\right)}+H_{2}O_{\left(g\right)},\;∆H_{700\;{}^{\circ}C}=52.086\;\mathrm{kJ}$$
(3)$$FeO_{\left(s\right)}+H_{2\left(g\right)}\to Fe_{\left(s\right)}+H_{2}O_{\left(g\right)},\;∆H_{700\;{}^{\circ}C}=14.488\;\mathrm{kJ}$$

From a thermodynamics point of view, chemical reactions (1) to (3) can occur at appropriate temperatures and H_2/_H_2_O ratios. For a typical industrial temperature of 700 °C, the H_2/_H_2_O ratio for chemical reactions (1) to (3) must be above 1 × 10^−5^, 0.7 and 2.8, respectively. In bauxite residue (BR; dewatered red mud), there is a significant amount of iron, with higher content than the utilized bauxite ore in the Bayer Process, and in addition, this material has some undigested alumina, about 15–25 wt% Al_2_O_3_ [11]. To further ensure eco-friendly metals and alumina production via bauxite residue valorization, the HARARE Project was started in 2021 and is a consortium of universities, research institutes and industrial companies from Norway, Greece, Germany and Belgium [21,22]. This project was established based on the ENSUREAL project achievements [23]. The main innovative approach is that the bauxite residue is agglomerated and then reduced with hydrogen. Furthermore, the metallic iron is separated by magnetic separation and alumina and rare earth elements are recovered via hydrometallurgical approaches. Hydrogen use for the reduction step ensures no direct CO_2_ emission from the process, as mostly H_2_O is emitted in this step. In a process scheme of the Harare project, lime is used as an additive to agglomerate bauxite residue and form more digestible compounds of aluminum, leachable calcium aluminates. Lime addition is also of great importance when it comes to inhibiting the leaching of silicon and titanium of BR, which increases the purity of alumina in later processing [24]. The present work presents the test of a new novel process on bauxite residue valorization by first hydrogen reduction, followed by alkaline leaching under specific conditions. No similar research on bauxite residue valorization was found in the literature to evaluate regarding the applied approach and obtained results.

## 2. Materials and Methods

### 2.1. Materials and Pelletizing

The experimental procedure for this paper includes the applied process steps, which are shown in the flowsheet represented in Figure 1, including the materials characterization techniques.

### 2.2. Sample Preparation and Pelletizing

Bauxite residue (BR) was provided by Mytilineos (Marousi, Greece) and the limestone (LS) was provided by Omya (Molde, Norway) as the starting raw materials. The XRF analysis of the raw material is presented in Table 1, Loss of ignition (LOI) data are also given in Table 1.

Both BR and LS were first deagglomerated and milled separately in a Retsch RS 200 ring mill. Then, 1000 g of BR and 640 g of LS was completely mixed in a turbulent mixer. The mixed powder was blended with water in a disc pelletizer until properly sized pellets (4–8 mm) could be extracted and further dried in an oven at 60 °C in 24 h. The dried pellets were sintered in six alumina boats in a muffle furnace (Nabertherm N17/HR Muffle Furnace) with a heating rate of 20 °C/min to the target temperature of 1200 °C, which was held for two hours. The furnace has a temperature gradient, so that the pellets never fully experienced 1200 °C inside the furnace. The pellets were left to cool down to room temperature in the furnace. In Table 2, the measured compositions of the pellets by XRF before and after sintering are presented.

### 2.3. Hydrogen Reduction

The H_2_ reduction experiments were carried out in a vertical tube resistance furnace. A schematic of the furnace is shown in Figure 2. The furnace is a vertical tube reactor built for this type of experiment.

Initially, a molybdenum crucible as sample holder was cleaned and weighed before the sample (25 g of sintered pellets) was added to the crucible. Thereafter, the crucible was mounted back into the furnace while gas inlets were connected to purge out any remaining oxygen with argon and helium. Purging with the inert gases was performed with 0.05 NL/min for 24 h; helium use after Ar purging was for safety issues and checking that it does not leak and that the furnace is safe for hydrogen use. The heating rate was 10 °C/min to the target temperature of 1032 °C, 1132 °C or 1232 °C under helium flow; hydrogen was then introduced with a gas flow of 0.2 NL/min, and the reduction lasted for 30, 60, 120 or 150 min. The temperature programs were set to 32 °C above the target temperature to compensate for the difference between wall temperature and the temperature profile within the crucible, so that the sample reaches 1000 °C, 1100 °C and 1200 °C. After each reduction experiment, the respective sample was weighed and recorded. In this study, nine experiments were carried out, and their details are presented in Table 3.

### 2.4. Leaching of Reduced Samples

Four reduced samples were selected and milled in a laboratory ring mill prior to leaching. They were samples that reduced for 1 h reductions at 1000 °C and 1100 °C, and for the ones reduced for 2.5 h, at 1000 °C and 1100 °C. About 5.0 g of samples was leached with 100 mL of a Na_2_CO_3_ solution (60 g/L) at 60 °C for 60 min in a beaker on a hot plate. After leaching, the mixture was filtered with a vacuum set-up until all the liquid had passed. The pore size of the filter used was 11 µm. The filter cakes were dried in an oven at 65 °C for 24 h before weighing. The filtrate for each sample was collected in containers, from which some of the filtrates of each sample (3 mL) were sent to for ICP-MS analysis (Agilent 8800 Triple Quadrupole ICP-MS). In addition to the final leachate samples, a blank sample of the Na_2_CO_3_-solution was taken and sent for ICP-MS as well. The dried filter cakes were collected in containers and were analyzed by XRD.

### 2.5. Characterization of Solid Materials

A Scanning Electron Microscope (SEM; Carl Zeis Microscopy GmbH, Jena, Germany) was used, and secondary electron imaging (SEI) and back-scattered electron imaging (BEI) were performed to investigate the microstructure of different samples. The SEM used was a JEOL Tabletop SEM model, and X-ray mapping was used to determine possible phases for later analysis. Point analysis was conducted to estimate possible phases in single spot from the coherent atomic relations of the elements. Additional mineralogical analysis of the materials and samples were carried out by using a Rigaku MiniFlex 600, with Cu-Kα radiation, in the 2-theta range of 10-to-90 degree diffraction angle and 0.02 degree step size.

## 3. Results

The results from the experimental work are summarized as follows.

### 3.1. Direct Observations

Measured weights and calculated values after sintering and after hydrogen reduction are presented in Table 4 and Table 5.

Average mass loss after sintering was 22.5%, which was higher than the calculated theoretical mass losses, calculated from Equation (4) for mass of CO_2_ removed via calcination of carbonates. The reduction experiment which yielded the most and least metallic iron, only accounting for direct observations, is reduction sample 4 and 7, respectively (Table 5). Reduction 4 was reduced for 2.5 h at 1000 °C and sample 7 was reduced at 0.5 h at 1200 °C.
(4)$$\frac{\frac{wt\%CaO}{100}\times m_{sample}}{M_{CaO}}\times M_{CO_{2}}=m_{CO_{2}}\;$$
where *M_CaO_* and $m_{CO_{2}}$ denote the molecular masses of lime (CaO) and carbon dioxide (CO_2_), while *m_sample_* and $m_{CO_{2}}$ denote the mass of the sample and carbon dioxide, respectively. Carbon dioxide is the main component with significant mass loss. The calculations for the theoretical total expected mass losses in Table 5 are described in the discussion chapter.

### 3.2. Phase Analysis

#### 3.2.1. Dried and Sintered Pellets

The XRD results of the dried pellets and the sintered pellets are compared in Figure 3 with the most possible phases identified.

In the dried pellets, we see calcite, hematite, diaspore and andradite as the dominant phases, while weak peaks of silica and titanium oxide were identified. In the sintered pellets, however, Gehlenite, brownmillerite, calcium titanate, calcium iron titanium oxide and silica were observed.

Figure 3 indicates that most of the phases existing in the dried pellets are changed upon sintering as volatile components are driven off, and more thermodynamically stable phases are formed via sintering at elevated temperatures.

#### 3.2.2. Reduced Samples

The XRD analyses of the reduced samples with the shortest and longest durations (0.5 and 2.5 h) and different temperatures (1000, 1100 and 1200 °C) are compared in Figure 4 and Figure 5.

It can be observed in Figure 4 that the iron oxides were completely reduced at 1000 °C, while there is some wüstite and less metallic iron in both the reduction at 1100 °C and 1200 °C. The same pattern applies for samples reduced for 2.5 h (Figure 5). In both short and long reduction durations at 1000 °C, the iron-containing phases were fully reduced to metallic iron, however, at 1100 °C and 1200 °C, the reduction is in lower extents. Comparing Figure 4 and Figure 5, it is observed that for a given temperature at 1100 °C and 1200 °C, reduction is more for the longer reduction time. However, the reduction is obviously faster at 1100 °C than 1200 °C.

#### 3.2.3. Leaching Residue Analysis

Results from the XRD analysis of the leaching residue for the reduced samples 4 and 7 are presented in Figure 6 and Figure 7, respectively. To easily compare with before leaching, the XRD spectrums of the corresponding reduced samples are re-presented in each figure.

Evaluating the XRD results in Figure 6 and Figure 7, it is obvious that the aluminum-rich phase, mayenite, has been highly leached. Moreover, the non-leached aluminum resides are the shulamitite and brownmillerite phases. All four analyzed leaching residue samples display a high intensity of iron and may indicate that iron is inert, and it is not dissolved and oxidized under the leaching conditions.

### 3.3. Microstructural Analysis

#### 3.3.1. Sintered Pellets

The overall structure and phase composition of each sample was studied in SEM at different magnifications. Both SEM imaging, X-ray mapping and EDS point analysis were performed to characterize the phases. For the sintered pellets, the composition seemingly consists of two dominant phases with a few white spots and large pores, as seen in Figure 8. In Figure 9, it is marked where the point analyses were taken.

Brownmillerite (Ca_2_ (Al, Fe^3+^) 2O_5_) and gehlenite (Ca_2_Al (AlSiO_7_)) represent the main phases, where brownmillerite is the brighter iron-rich phase and gehlenite is the darker phase and contains larger amounts of aluminum (Figure 8). The relative composition of each phase can deviate from the general formula, as different elements can be dissolved in the structure of these phases. The X-ray mapping of elements indicates that the selected elements of interest are widely distributed, and the phases are relatively complex. However, it can be observed that the brighter phases have an abundance of iron, which indicates the iron-rich phase brownmillerite. Aluminum is in areas without iron but in association with silicon, which suggests the presence of gehlenite in the darker areas. Calcium is distributed through all phases of the sintered pellet as expected due to lime addition.

#### 3.3.2. Reduced Samples

As for the sintered pellet, the reduced pellets were characterized with SEM imaging, X-ray mapping of elements and EDS point analysis. In the reduced samples 1 (1000 °C, 0.5 h) and 6 (1100 °C, 1 h), the structure of the pellets changed from the sintered pellet. It is seen in Figure 10 that iron particles started to join and agglomerate, creating larger iron grains. At 1000 °C, the phases have more clear boundaries, whereas for the reductions at 1100 °C, the phases are more mixed, (Figure 10).

The areas of interest were analyzed with points analysis, as shown in Figure 11, for a typical sample. Reduced sample 1 (1000 °C, 0.5 h) contains three main phases: metallic iron (Fe), mayenite (12CaO∙7Al_2_O_3_) and larnite (Ca_2_ (SiO_4_)). It is seen in Figure 11 that the iron particles are reduced and start to form larger grains. Other elements such as aluminum and calcium constitute phases such as mayenite and larnite. At the boundaries of the iron agglomerates, oxygen can be spotted along with calcium, and minor amounts of aluminum and titanium, which indicates the presence of shulamitite (Ca_3_Ti (Fe_1.18_ Al_0.82_) O_8_).

### 3.4. Leachate Composition

The measured concentrations of aluminum, silicon and iron in the leachate solutions are given in Table 6. As seen, the obtained solutions contain significantly high concentrations of Al, while Si is much lower, and Fe is insignificant. Comparing the leachates with the used leaching agent (blank sample), it is obvious that Al, Si and Fe all come from the reduced samples.

## 4. Discussion

### 4.1. Evolution of Phases in Sintering

When comparing the LOI in the XRF results of dried pellets (Table 2) and the mass loss after sintering (Table 4), there is slightly higher mass loss in sintering than the measured LOI. This may be due to the application of higher temperature and duration in the sintering process than LOI measurement, which is at about 900 °C for shorter time. The main mass loss in sintering is obviously due to the decomposition of calcite (CaCO_3_ → CaO + CO_2_), which is about 17% of the mass loss. Observing higher mass loss in sintering than the calculations in Table 3 can be due to the decomposition of diaspore that exists regarding the XRD pattern of the dried pellets (Figure 3) and other minor hydroxides in the pellets. These hydroxides are decomposed at lower temperatures than the calcite decomposition.

During the sintering experiments, the main phases in the dried pellets are changed to calcium-containing phases, as free CaO is accessible to the BR components to react with, which can be seen in Figure 3. Complex reactions form new, more thermodynamically stable phases, suitable for the reduction experiments. The iron content from hematite and andradite is redistributed in brownmillerite and calcium–iron–titanium oxide. The bound aluminum from diaspore and andradite also changes to new calcium-rich phases brownmillerite and gehlenite. The andradite phase, which contains many elements, is likely distributed evenly between the product phases from the sintering. The titanium oxide of BR forms new phases with calcium and iron as calcium–titanium oxide and calcium–iron–titanium oxide. Lastly, the silicon remains as silica in the sintered pellets, as well as existing in the gehlenite phase.

### 4.2. Reducibility of Pellets by Hydrogen

To enable the comparison of different reductions, and examine their kinetics, a fraction conversion (X) from hematite to metallic iron was calculated by Equation (5) [14].
(5)$$X=\frac{∆m}{\left(\frac{wt\%Fe_{2}O_{3}}{100}\right)\times \left(\frac{3*M_{O}}{M_{Fe_{2}O_{3}}}\right)\times m_{i}}$$
where ∆*m* denotes the mass change during reduction, *m_i_* denotes the initial sample mass and *wt%Fe*_2_*O*_3_ denotes the *wt%* of hematite in the initial sample from Table 2. *M_O_* denotes the atomic weight of oxygen (16.0 g/mol), and $M_{Fe_{2}O_{3}}$ denotes the molecular weight of hematite (159.7 g/mol) [25].

The graph shown in Figure 12 indicates the importance of both time and temperature on the yield of metallic iron from Table 5, where temperature seems to be of the greatest importance. When comparing the overall yield of the different reduction experiments (Table 5), the reduction with the highest iron yield is reduction 4, with 82.9%, while the highest yield at 1100 °C is reduction 7, with 70.5%. This further indicates that time is increasingly important, especially at higher temperatures. Considering the Fe-containing phases in the sintered pellet (non-stoichiometric hematite, brownmillerite and Calcium-iron-titanate), the amount of oxygen in the material in association with Fe is less than that in Fe_2_O_3,_ which was considered in Equation (6). Hence, the actual reduction extent for the different times and temperatures is slightly higher than those seen in Figure 12. However, it is challenging to precisely calculate these reduction extents, as the quantity of Fe-containing phases in the sintered pellets could not be determined.

#### 4.2.1. Effect of Temperature

The overall characteristics of the reduction experiments indicate that the applied temperature at 1000 °C is the most reliable temperature to reduce iron-containing compounds. Based on the XRD results, the experiments carried out at 1000 °C were accompanied with complete reduction, while the only exception being reduction experiment 1 (1000 °C, 0.5 h), which indicated incomplete reduction, and a small amount of the brownmillerite phase remains. In addition, point analysis of reduction suggests that both brownmillerite and shulamitite are present in reduction 1. It is worth mentioning that these two phases were also detected in the reduced sample 4, and the small amount of them was not observed in the corresponding XRD spectrum in Figure 5.

The reduction experiments at 1100 °C show a more complex reactive system. At 1100 °C, new phases exist, as seen in the XRD results (Figure 4 and Figure 5); brownmillerite and wüstite are present in both durations 0.5 and 2.5 h. These phases indicate incomplete reduction of iron compounds and are most present in the shortest experiment, reduction 5 (0.5 h). Kinetics and thermodynamics suggest that reduction reaction rate should increase with temperature [20,26]. However, SEM images of reduction experiment 6 (Figure 10) show that the sample has a semi-molten phase throughout the pellet. This phase seemingly covers up small pores, inhibiting mass transportation of reactants such as gas and products, limiting the overall rate of the reduction reaction. With a less porous structure, the pore diffusion decreases, and solid-state diffusion governs and become the rate-limiting step of the reaction, thus demanding more time to reduce the sample [26].

The experiments carried out at 1200 °C show no remarkable reduction to metallic iron. These samples were fully melted in the crucible during the experiments. The XRD analysis of the samples (Figure 4 and Figure 5) shows that iron compounds are barely reduced, only leaving a weak diffraction peak for iron, while a relatively strong peak of wüstite is observed. Since the pellets were almost melted, the pores most likely dissipated, and hydrogen gas diffusion into the material was greatly inhibited, causing negligible reduction—hence the reduction occurs mostly at the external surface of the molten material exposed to hydrogen. This melting effect practically prevents any further reduction of the phases containing iron in the melted pellets.

#### 4.2.2. Effect of Reduction Duration

The comparison of the different reduction durations revealed that longer intervals were beneficial for higher iron recovery. XRD results for the reduction experiments at 1000 °C show the almost complete reduction of iron compounds even at the shortest time of 0.5 h. Although a small amount of brownmillerite, wüstite and shulamitite is left, however, they were reduced in longer durations. This may indicate that the rate of gas–solid reactions is high for the porous pellet at 1000 °C. The reduction rate at 1100 °C is obviously more dependent on time, and the XRD analysis of reduction at 1100 °C (Figure 4) shows the appearance of more wüstite and brownmillerite than at 1000 °C. These phases indicate incomplete reduction of iron compounds to FeO as seen for the reduced sample 5 within 0.5 h. Longer durations at this temperature show that the amount of intermediate FeO phases is decreased, and more iron is formed. For the reductions at 1200 °C, which are semi-molten pellet, the duration of the experiments had an insignificant effect on the further reduction of FeO. It is conceivable that experiments at this temperature can become sufficiently reduced given enough time. The theoretical and experimental mass loss from the reduction experiments, considering iron, is presented in Table 5. The respective reduction extents for the experiments at the same temperatures but different times confirm the XRD graphs’ indication that the metallic iron concentration increases with time.

#### 4.2.3. Pellet Physical Properties Effect

Considering the effect of both temperature that affects the physical properties of pellets and the duration of reduction, the reduction phenomena can be more precisely explained. At lower temperatures, i.e., 1000 °C, the porous structure is not lost, the rate of reduction is high and the reactant hydrogen diffuses rapidly into the pellets, and the chemical reactions are governed. However, at higher temperatures, the phases in the pellets start to soften and melt, and they are accompanied with decreasing pellet porosity; the reaction rate is decreased. The formation of metallic iron in all temperatures indicates that it is the most stable phase, and all Fe-containing phases must be reduced to metallic Fe. However, at higher temperatures of 1100 °C and 1200 °C, the reduction to FeO was significant, while further reduction of FeO to Fe was slow. Hence, we may conclude that the last step of reduction beyond FeO from a semi-molten medium by H_2_ gas is a rate-controlling step at the higher applied temperatures.

To achieve optimal reducibility of the samples, the structure of the pellet is as important as the experimental parameters. Using hydrogen gas as the reactant, H_2_ molecules must be well-transported and achieve good contact with the iron-containing phases in the pellet to reduce them and yield metallic iron. The mass transports of H_2_ gas and H_2_O product gas are highly dependent on pellet porosity, and enough open pores are needed for the required penetration/diffusion of these gases into the pellet. Obviously, applying higher temperatures than 1000 °C causes porosity loss during the reduction as semi-molten phases are formed, as the SEM investigation indicates that the different phases seemingly start to melt. The partial or complete melting of some phases causes the decrease in open porosity, and it limits the mass transport of H_2_ gas into the pellets, and it will slow down the overall reaction rate. When the reaction rate decreases, the duration of the experiment becomes important. These observations, and the fact that the rate of chemical reduction reactions is in principle higher at higher temperatures, can conclude that the reduction process is mass transport controlled (H_2_ gas transport).

### 4.3. Leaching Behavior of Reduced Pellets

The results of the leaching experiments on the above-mentioned four reduced samples are discussed as follows.

#### 4.3.1. Characteristics of Leaching Residue

Because of the predominance of iron in the leaching residue samples, their XRD analyses (Figure 6 and Figure 7) show high intensity of iron peaks. The reactions from leaching of various calcium–aluminate phases are yet to be determined, as the compound can be found in multiple stoichiometric relationships, where most of them result in the leaching of alumina [9]. However, the major and desired phase is mayenite, considering its ability to be digested [7]. The XRD analyses show a significant presence of mayenite after reduction and that the phase is completely dissolved in leaching. The digesting reaction for mayenite (12CaO∙7Al_2_O_3_) is shown in Equation (6) and it has been discussed in the literature [7,13]. However, there is an increase in the aluminum-containing phase, shulamitite, in the leaching residues. This indicates that most of the alumina in other phases has been dissolved, whereas the rest remains in a non-leachable shulamitite phase. The stability of this phase is an advantage for the alumina recovery as it does not contaminate the produced sodium aluminate solution and hence the precipitated hydrated alumina.
(6)$$12CaO\cdot 7Al_{2}O_{3\left(s\right)}+12Na_{2}CO_{3\left(aq\right)}+5H_{2}O_{\left(l\right)}\to 14NaAlO_{2\left(aq\right)}+12CaCO_{3\left(s\right)}+10NaOH_{\left(aq\right)}$$

Gehlenite was not observed in the reduced samples at 1000 °C for 2.5 h, however, it is present in the solid materials both prior and post leaching for the reduced sample 7, which substantiates the statement that it is non-leachable [5]. The gehlenite peak at approximately 29 degrees is presumed to still be present in the residue, although it is overlapping with the calcite peak identified at the same degree. Furthermore, the samples show high intensity of the calcium-containing phases calcite and vaterite, as expected previously [7]. Other calcium-containing phases present both before and after leaching are larnite and calcium titanate, indicating that they are less-to-non- leachable. The smaller yield of alumina for samples 6 and 7 correlates with the presence of brownmillerite in these samples (XRD in Figure 7), indicating that it is not leachable, and it is a stable phase in the applied leaching conditions. Considering the non-leachability of shulamitite and brownmillerite phases and the fact that they contain both Fe and Al, it is best to have a high degree of H_2_ reduction, so that Fe is more recovered, and in parallel, less Al is lost in the leaching process.

#### 4.3.2. Alumina Recovery

The aluminum recovery was calculated regarding theoretical mass balances, calculated from XRF data presented in Table 2. Equation (7) was used, while results can be found in Table 7.
(7)$$Aluminium\;yield=\frac{(C_{Al,\;sample}-C_{Al,\;blank})\times V_{sample}}{\left(\frac{2\times M_{Al}}{M_{Al_{2}O_{3}}}\right)\times \frac{wt\%Al_{2}O_{3}}{100}\times m_{sample}}\times 100\%$$
where *C_Al,sample_* denotes the concentration of aluminum present in the sample from ICP-MS analysis, *C_Al,blank_* denotes the aluminum concentration obtained from the blank (solution used for leaching) and *V* denotes the volume of the sample in leaching. *M_Al_* denotes the atomic weight of aluminum (27.0 g/mol), *M_Al_*_2*O*3_ denotes the molecular weight of *Al*_2_*O*_3_ (102.0 g/mol) and *m_sample_* denotes the initial mass of the leached sample [25]. The *wt% Al*_2_*O*_3_ is from the XRF results of the sintered pellets in Table 2. The Si yields to the solution in Table 7 were calculated in a similar manner as Al.

From this, it was found that aluminum yield into the solution decreased with increasing reduction temperature and time, as reduced sample 2 yielded the most Al. The higher Al recovery for the samples at lower temperatures and the same reduction durations is attributed to having lower amounts of the non-leachable Al-containing phases of gehlenite and brownmillerite. Observing a lower Al recovery for the samples with the same reduction temperature and longer reduction time in reduction test 2 compared with 4, and 6 compared with 7, is complicated to be discussed. A possibility of obtaining lower Al recovery at longer reduction times with close reduction degree and quantities of leachable and non-leachable phases may be obtaining lower porosity at longer high-temperature treatment. This is due to the sintering phenomenon in pellets and, correspondingly, obtaining lower porosity and surface area for the crushed reduced pellets. Lower porosity may decrease slower solution penetration or faster passivation via calcite layer formation, as has been described previously [8].

#### 4.3.3. Silica Dissolution in Leaching

The ICP-MS analysis of the solution (Table 6) indicates that small amounts of silica have been digested, about 4% regarding Table 7 [13]. The XRD analyses show the presence of the silicon-containing phases larnite and gehlenite both prior and post leaching in the residue. As gehlenite is considered a non-leachable phase [5], it is suspected that the dissolved silica most likely originates from contaminations in mayenite, as this phase has been dissolved completely, and it has small amount of Si in its crystal structure [27]. The larnite phase is abundant both prior and post leaching, indicating that it is not highly leachable, and it cannot be a source for Si contamination.

## 5. Conclusions

Oxide-sintered pellets were made from a mixture of bauxite residue and calcite. The pellets were reduced by hydrogen gas at elevated temperatures, and selected samples were leached by a sodium carbonate solution. The main conclusions are summarized as:In the pelletizing and sintering process iron- and Al-containing phases in raw materials are converted mainly to brown millerite and gehlenite, respectively.In hydrogen reduction at elevated temperatures, metallic iron is formed from brownmillerite and calcium–iron–titanate phases via the intermediate wüstite phase formation.The hydrogen reduction at 1000 °C occurs as a gas–solid reaction, while at higher temperatures it is changed to a gas–solid–liquid system due to the partial melting of pellet components.The rate and extent of hydrogen reduction was highest for 1000 °C and the lowest for 1200 °C. The process is more chemical-reaction-controlled for the former, while it is diffusion-controlled for the latter.During hydrogen reduction, gehlenite is converted to a leachable mayenite phase.Al recovery via the alkaline leaching of reduced pellets with sodium carbonate solutions yields high Al recovery (up to 87% measured) with no effect on metallic iron, while calcite is evolved in the leaching residue.

## Figures and Tables

**Figure 1 materials-15-06012-f001:**
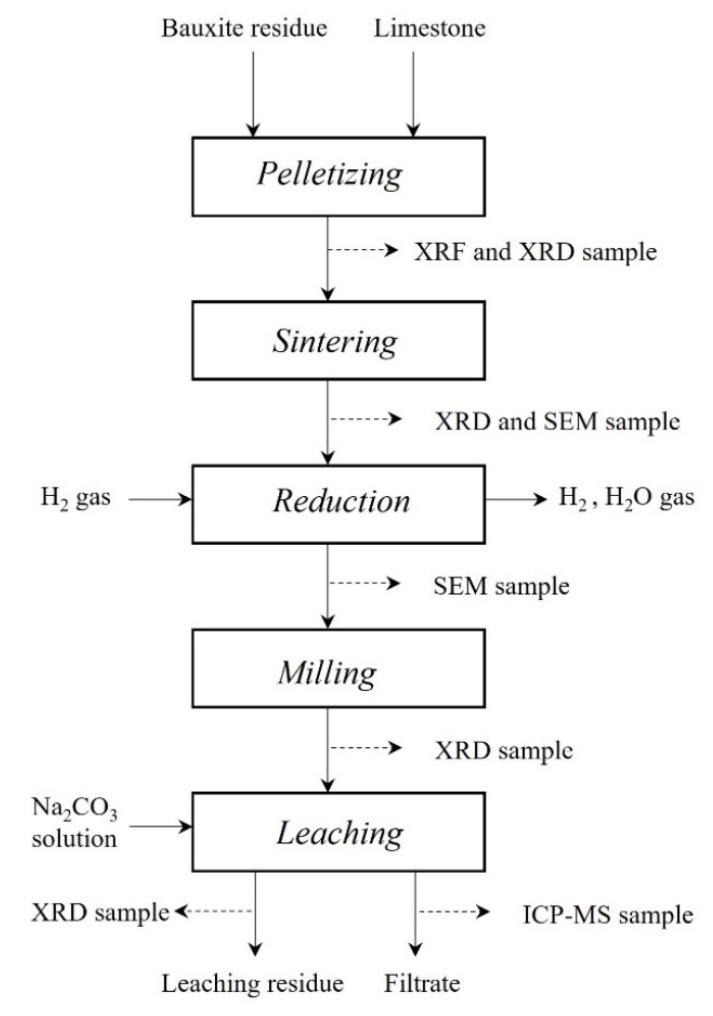
A schematic overview of the experimental procedure and applied characterization techniques.

**Figure 2 materials-15-06012-f002:**
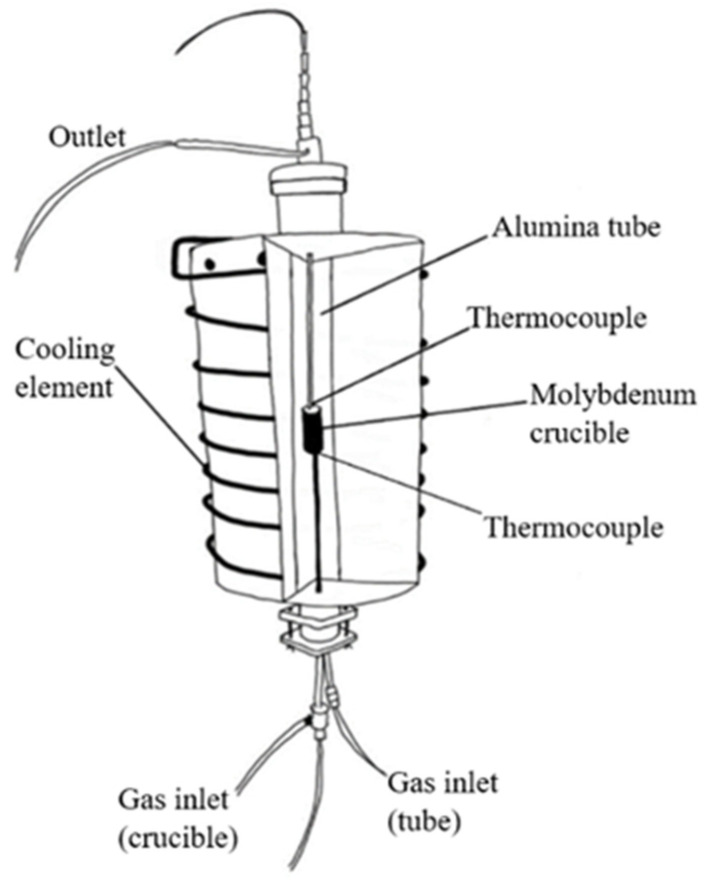
A schematic drawing of the reduction furnace.

**Figure 3 materials-15-06012-f003:**
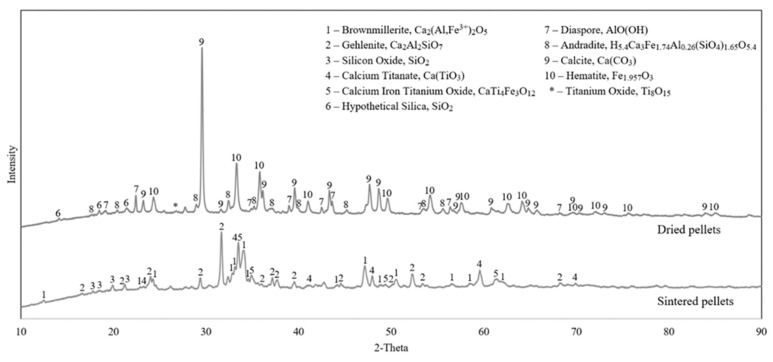
XRD analyses of the dried pellets (**top**) and sintered pellets (**bottom**).

**Figure 4 materials-15-06012-f004:**
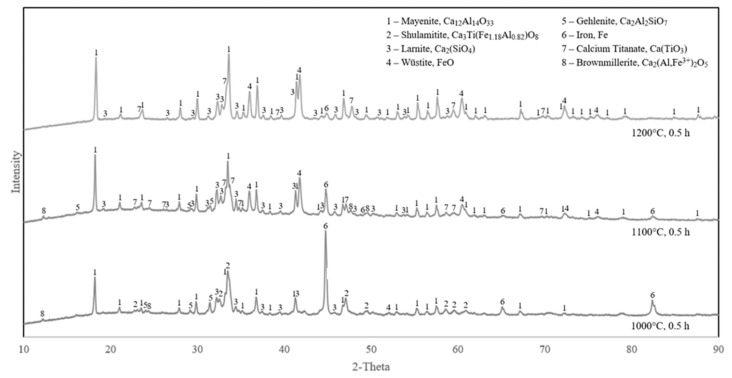
XRD analyses of samples reduced for 0.5 h at different temperatures (samples 1, 5, 8).

**Figure 5 materials-15-06012-f005:**
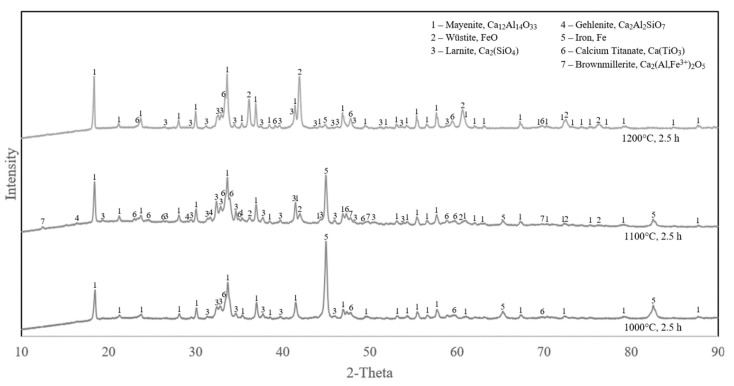
XRD analyses of samples reduced for 2.5 h at different temperatures (Samples 4, 7, 9).

**Figure 6 materials-15-06012-f006:**
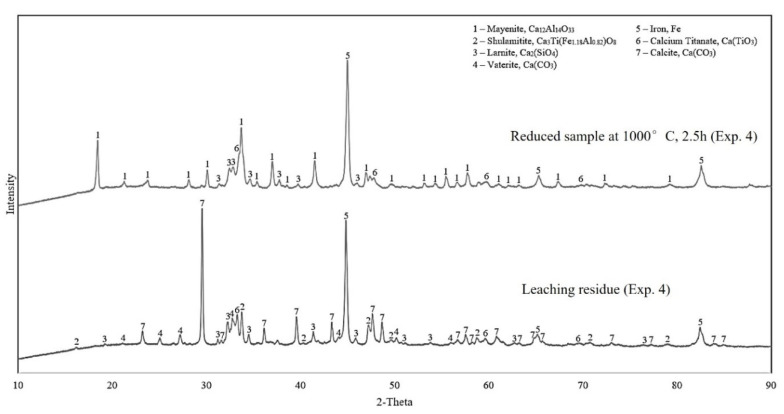
The XRD spectrums of leaching residue of the reduced sample 4 in comparison with the solid before leaching.

**Figure 7 materials-15-06012-f007:**
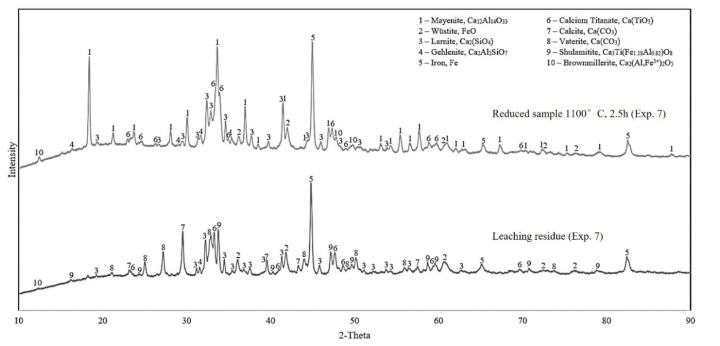
The XRD spectrums of leaching residue of the reduced sample 7 in comparison with the solid before leaching.

**Figure 8 materials-15-06012-f008:**
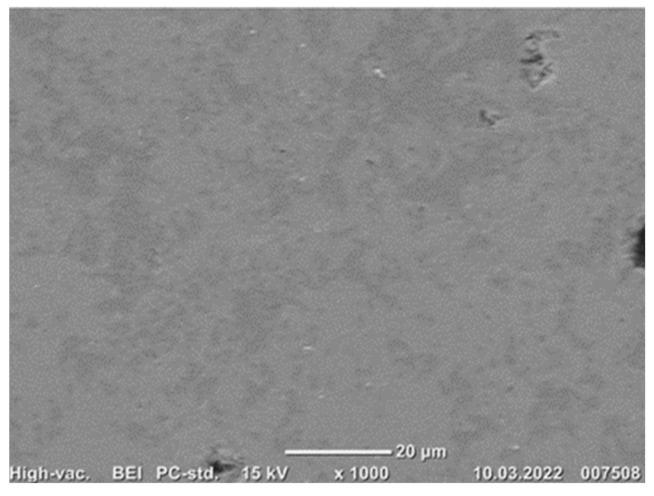
Typical SEM-BEI of sintered pellet.

**Figure 9 materials-15-06012-f009:**
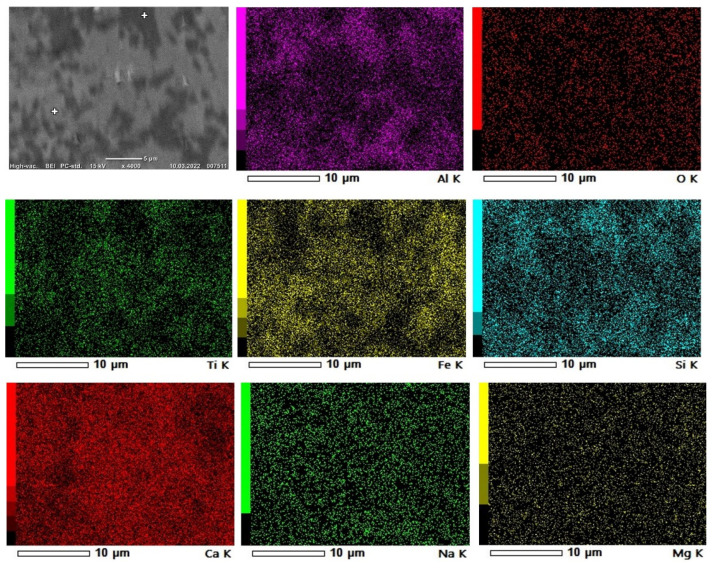
X-ray mapping and BEI of sintered pellet.

**Figure 10 materials-15-06012-f010:**
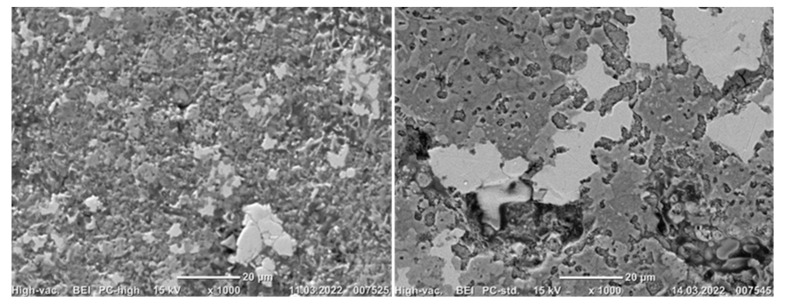
BEI imaging of reduced samples 1 (**left**) and 6. (**right**).

**Figure 11 materials-15-06012-f011:**
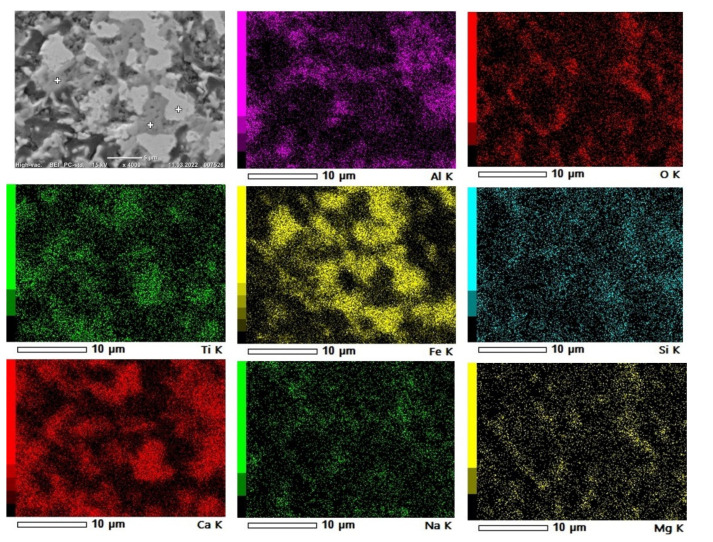
X-ray mapping of elements and BEI of reduced sample 1.

**Figure 12 materials-15-06012-f012:**
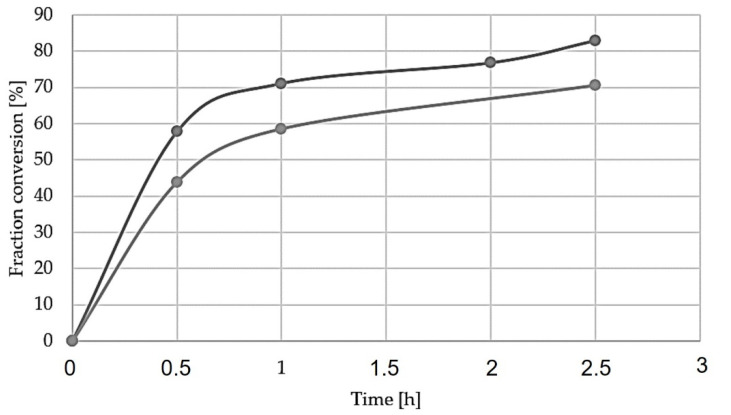
Fraction conversion of hematite to metallic iron as a function of time.

**Table 1 materials-15-06012-t001:** XRF analysis of raw materials; limestone (LS) and bauxite residue (BR).

Materials	Al_2_O_3_	CaO	Fe	Fe_2_O_3_	K_2_O	MgO	MnO	Na_2_O	P	P_2_O_5_	S	SO_3_	SiO_2_	TiO_2_	LOI
LS [wt%]	0.90	52.70	-	0.15	0.12	0.95	-	-	-	0.01	-	0.06	2.07	0.03	42.60
BR [wt%]	22.00	8.80	28.50	-	0.09	0.23	0.08	3.10	0.05	-	0.38	-	7.10	5.00	24.67

**Table 2 materials-15-06012-t002:** XRF results for dried pellets (DP) and sintered pellets (SP).

Materials	Al_2_O_3_	CaO	Cr_2_O_3_	Fe_2_O_3_	K_2_O	MgO	MnO	Na_2_O	P	S	SiO_2_	SrO	TiO_2_	V_2_O_5_	ZrO_2_	LOI
DP [wt%]	14.60	27.70	0.11	27.07	0.09	0.48	0.09	2.09	0.03	0.23	5.42	0.09	3.25	0.09	0.11	18.56
SP [wt%]	18.15	34.34	-	33.12	0.05	0.62	0.10	2.05	0.04	0.31	6.77	0.10	4.10	0.10	0.10	0.07

**Table 3 materials-15-06012-t003:** Experimental conditions during hydrogen reduction and initial weight of each sample.

Temperature [°C]	1000			1100			1200		
Hydrogen gas flow	0.2 NL/min								
Time [h]	0.5	1	2	2.5	0.5	1	2.5	0.5	2.5
Reduction sample	1	2	3	4	5	6	7	8	9
Sample weight [g]	25.0	25.2	25.1	25.1	25.0	25.1	25.7	24.9	24.1

**Table 4 materials-15-06012-t004:** Initial weight and mass losses during sintering.

Initial Weight [g]	Mass Loss [%]	Theoretical Mass Loss [g]	Actual: Theoretical Mass Loss [%]
82.6	22.7	17.9	104.8
84.2	22.5	18.3	103.9
85.6	22.3	18.6	102.7
76.0	22.5	16.5	103.0
88.7	22.5	19.3	103.8
78.5	22.7	17.1	104.4
Average:	22.5	17.9	103.8

**Table 5 materials-15-06012-t005:** Mass loss, theoretical mass loss and Fe yield after hydrogen reduction of pellets.

Reduction No.	Mass Loss [g]	Mass Loss [%]	Theoretical Mass Loss [g]	Fe Yield [%]
1	1.44	5.75	2.49	57.8
2	1.78	7.07	2.50	71.1
3	1.92	7.65	2.50	76.8
4	2.07	8.27	2.50	82.9
5	1.09	4.34	2.49	43.8
6	1.46	5.84	2.50	58.5
7	1.8	7.02	2.55	70.5
8	0.73	3.04	2.40	30.5

**Table 6 materials-15-06012-t006:** Concentration of Al, Si and Fe in leached samples, analyzed by ICP-MS.

Sample	Concentration [mg/L]		
	Al	Si	Fe
Blank	0.12	3.78	0.00
Reduction 2	4163	70	0.78
Reduction 4	3943	66	0.74
Reduction 6	3915	64	0.68
Reduction 7	3647	66	0.55

**Table 7 materials-15-06012-t007:** Aluminum and silicon yield into the leaching solution.

Sample	Yield [%]	
	Al	Si
Reduction 2	86.6	4.40
Reduction 4	82.0	4.14
Reduction 6	81.5	4.07
Reduction 7	75.9	4.14

## Data Availability

Data available in a publicly accessible repository that does not issue DOIs.

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
