# Peer review of "Isothermal Hydrogen Reduction of a Lime-Added Bauxite Residue Agglomerate at Elevated Temperatures for Iron and Alumina Recovery"

_materials, 2022, doi:10.3390/ma15176012_

Round 1

Reviewer 1 Report

The paper presents an interesting topic, but the work has little significance as the experiments are poorly backed up by the analysis and theoretical explanations. For example, the authors attempt to tackle thermodynamics by presenting enthalpy for iron oxide reduction with hydrogen at 700 °C. This is a wrong approach from a metallurgical view and a gross simplification of the investigated processes. The whole analysis of the process is limited to mass change and XRD analysis. The XRF and SEM analysis is weak at best. There is some additional leaching experiments, but the relevance is not explained or presented. The paper simply lacks a scientific approach and offers little to no new or even established insights on the reactions occurring during the reduction.

Author Response

Thank you for the comments and sharing your points!

The hydrogen reduction of iron oxides has been studied in literature through many studies, and the typical applied temperature of 700C in section 1 for chemical reactions (1) to (3) is only to show the magnitude of enthalpy changes. There is no intend to tackle thermodynamics, the reaction is possible in a wide temperature range with hundreds publications and pilot/industrial plants in operation. To clarify, the following text was added in the introduction in lines 73-76 in revised manuscript:

“From thermodynamics point of view chemical reactions (1) to (3) can occur at appropriate temperatures and H2/H2O ratios. For a typical industrial temperature of 700°C, the H2/H2O ratio for chemical reactions (1) to (3) must be above 1×10-5, 0.7 and 2.8, respectively.”

The applied characterization methods in the research were through XRF, XRD, ICP-MS analysis and SEM investigations, supported by mass changes studies. These are common techniques that are used to do research in the field of extractive metallurgy. Thermogravimetry and Differential Thermal Analysis (TGA/DTA) are other techniques that we currently use, and the results will be published in later publications of the same project. The manuscript is about a new process and the first publication in a large research EU project, and it is not expected that it contains everything that a reader expects, the paper is already large with significant results and  new data. In the later publications of this project, we will apply other approaches and methodologies to discover more about the hydrogen reduction of bauxite residue agglomerates. Unfortunately, it is not clear that what is missing in SEM and XRF results that the reviewer mentions that they are weak results. The XRF analysis was carried out by a company (see section 2) that we have worked many years with, and we are quite satisfied from their measurements. The reason of doing leaching experiments after the reduction was to recover alumina into a sodium-aluminate solution (for further alumina recovery), and this is mentioned in the last paragraph (lines 83-95) of section 1 in the revised manuscript.

Reviewer 2 Report

The paper is well structured; the topic studied, the methodology used and the results are explained very well. The analytical techniques are adequate and the description of the sample preparation is well detailed. Consequently, from the point of view of the organization, it can be published in the form presented.

 The authors would be asked if it is possible to calculate an energy balance in order to evaluate the energy required for this type of treatment in order to complete the work also from this point of view.

 It is also advisable to check the formatting of the paragraphs, paying attention to the spaces between figures and text that are not always uniform within the paper.

Author Response

The positive comments from the reviewer are appreciated!

The manuscript is about a new process and it is the first publication in a large research EU project. The manuscript is already large with significant results and new data. We are working do energy and mass balances of the process and in our next publications of this project.

Reviewer 3 Report

The paper is not complete, even if it could potentially be a good work. It needs to be improved with following revisions:

1) The objective, methodology, and results should be better described, discussed and justified.

2)
 The results should be expanded significantly and quantitatively.

3) I strongly suggest that authors shall carry out more studies to compare the results from this paper to that from other similar studies.

Author Response

Thank you very much for the comments! The following actions were made to improve the manuscript:

1. The last paragraph in the section 1 was modified and improved to provide better description and justification.

2. The result section provides the main obtained results in the research, and they could not be expanded more. It is possible to add more results such as more SEM and XRD analysis data, however, no more new information that has been already presented in the results section will be introduced.

3. The performed research work on the hydrogen reduction of bauxite residue is the first publication in the field of bauxite residue valorization in HARARE EU project (via limestone addition) and we could not find similar previous works (on similar raw material, methodology, and using hydrogen) to compare with our research results.

Reviewer 4 Report

This research mainly presents a method to recover iron and alumina from the bauxite residue in a new process route. The research content is sufficient and has strong applicability, and the organizational structure is well. However, this paper needs a minor revision as suggested below.

1. The innovation of the article should be further condensed and extracted.

2. SEM, SEI and BEI tests should provide specific model number and test parameters.

3. The readability of the full manuscript should be further enhanced, because the audience may be not for related majors. When using abbreviations such as LOI, the full name should be presented when it appears at the first time, etc.

4. Everyone has different writing habits, I think it is better that the results and discussion can be combined. When I read the discussion, I have to find the corresponding results, which will weaken the coherence and readability of the whole manuscript.

5. English language and style are fine/minor spell check required.

Author Response

Thanks for the comments, they were constructive! The following actions were made to improve the manuscript:

  1. The innovation of the work is clearly described in the last paragraph in the section 1 of the revised manuscript.
  2. The whole manuscript was revised and for the SEM, SEI and BEI tests more information were added. The operational conditions of SEM are given in the SEM images, and information about the SEM equipment in section2.
  3. The whole manuscript was improved with providing information for example the full name for LOI, which is “Loss of Ignition” was added in line 107 of the revised manuscript.
  4. We agree to some extent with the reviewer comment about combining the results and discussion sections. However, we used the template that has been already designed for the “Materials Journal”. On the other hand, we have experienced that many reviewers ask for separate results and discussion sections.
  5. The whole manuscript was improved regarding grammar and spell check.